# In-Hospital Mortality in Non-COVID-19-Related Diseases before and during the Pandemic: A Regional Retrospective Study

**DOI:** 10.3390/ijerph182010886

**Published:** 2021-10-16

**Authors:** Nicola Bartolomeo, Massimo Giotta, Paolo Trerotoli

**Affiliations:** Department of Biomedical Sciences and Human Oncology, University of Bari Aldo Moro, 70124 Bari, Italy; nicola.bartolomeo@uniba.it (N.B.); massimo.giotta@uniba.it (M.G.)

**Keywords:** COVID-19, in-hospital mortality, comorbidity, hospital discharge record

## Abstract

Italy was one of the nations most affected by SARS-CoV-2. During the pandemic period, the national government approved some restrictions to reduce diffusion of the virus. We aimed to evaluate changes in in-hospital mortality and its possible relation with patient comorbidities and different restrictive public health measures adopted during the 2020 pandemic period. We analyzed the hospital discharge records of inpatients from public and private hospitals in Apulia (Southern Italy) from 1 January 2019 to 31 December 2020. The study period was divided into four phases according to administrative restriction. The possible association between in-hospital deaths, hospitalization period, and covariates such as age group, sex, Charlson comorbidity index (CCI) class, and length of hospitalization stay (LoS) class was evaluated using a multivariable logistic regression model. The risk of death was slightly higher in men than in women (OR 1.04, 95% CI: 1.01–1.07) and was lower for every age group below the >75 years age group. The risk of in-hospital death was lower for hospitalizations with a lower CCI score. In summary, our analysis shows a possible association between in-hospital mortality in non-COVID-19-related diseases and restrictive measures of public health. The risk of hospital death increased during the lockdown period.

## 1. Introduction

In 2019, more than 7,000,000 patients were discharged from Italian hospitals, of which more than 75% were public hospitals. The most frequent cause of hospitalization was cardiovascular disease, accounting for 14% of all discharges. Urinary bladder, breast, and lung cancer were the most frequent cancers, accounting for at least 3% of causes of hospital discharge. In 2019, discharges from Apulia accounted for 5.7% of all Italian discharges. Of these, 92.4% were in-hospital patients and 7.6% were day-hospital discharges. The hospitalization rate had been decreasing since 2010, from 115.8 per 1000 inhabitants to 90.1 per 1000 in 2019 [1].

Italy is the European nation most affected by the severe acute respiratory syndrome coronavirus 2 (SARS-CoV-2) infection, beginning in February 2020 [2,3]. The first case of SARS-CoV-2 infection in Italy was registered in Codogno on 19 February 2020; two days later, on 21 February 2020, the first death with SARS-CoV-2 provided as cause was registered in Vo’ Euganeo [4]. Due to the rapidly increasing cumulative incidence of coronavirus disease 2019 (COVID-19) in the European Union/European Economic Area and the United Kingdom, a short window of opportunity was indicated during which countries had the possibility to further increase their control efforts to slow down the spread of SARS-CoV-2 infection and decrease the pressure on their healthcare systems [5]. On 23 February 2020, the Italian government approved a localized lockdown measure to prevent the spread of SARS-CoV-2 infection [6]. On 9 March 2020, the Italian government imposed a national lockdown and approved several public health measures: quarantine for infected patients, social distancing, use of a surgical mask, limitation of individual freedoms, closure of commercial activities, closure of schools, travel restrictions, and reduction of displacements [7,8,9]. The Italian Ministry of Health recommended that each region should increase the hospital beds available for COVID-19 patients in the National Health Service (SSN, Servizio Sanitario Nazionale) [10] and manage other hospital activities that can be deferred due to the pandemic [11,12]. This strategy caused delayed access to healthcare, diagnostic procedures, and oncological screenings [13].

It could be argued that a reduction of hospital care has occurred. As a consequence, patients in the hospital setting could be different subjects for severity, type of disease, stage of disease and other characteristic, respect to those before the pandemic. Research by other authors has shown that health outcomes could be affected by these alterations of the treatment setting. In the USA, it was observed that online searches for oncological diseases and related screening had a sharp decline from the beginning of the COVID-19 pandemic [14]. During the lockdown period, a general reduction in hospital access for non-COVID-19 patients was observed; hospitalization was reduced for both medical and surgical cases and for conditions requiring intensive care support [15]. In emergency departments (EDs), it was reported that the rate of admission for critical patients was inversely related to the rate of admission for non-critical patients [16]. A clear reduction in hospitalizations for heart diseases during the beginning of the COVID-19 pandemic was observed [17]; the reduced number of hospital admissions has resulted in an increase in out-of-hospital deaths and long-term complications [18]. In a French study, a numerical increase in the rate of in-hospital deaths during the lockdown was observed; the increase was less than that found in an Iranian set [19].

These specific hospital mortality data could be useful for assessing the quality of healthcare and health systems. Measures could be used for benchmarking, particularly in conditions where case-specific mortality may be informative, to contrast mortality across hospitals, to examine the relationship between hospital mortality and variation in health system inputs [20]. In-hospital mortality was defined as death occurring during hospital stay.

In this retrospective study, the hospital mortality rate of non-COVID-19 patients in 2020 was compared with that in 2019. The first aim was to evaluate changes in in-hospital mortality and their possible relation with different restrictive public health measures adopted during the 2020 pandemic period compared to the 2019 pre-pandemic period. The secondary aim was to analyze the effects of patient comorbidities and health restrictions measures on in-hospital mortality.

## 2. Materials and Methods

We conducted a retrospective observational study using anonymous aggregated data extracted from the healthcare administrative databases of the southern region of Italy, Puglia. Regional data were collected and stored in the Regional Information System. Access to data is regulated by a regional policy to allow the use of data for epidemiological purposes to Regional Epidemiological Observatory. Data were provided after anonymization and treated according to current laws for treatment of health data. We analyzed the hospital discharge records (HDRs) of inpatients from public and private hospitals in Puglia from 1 January 2019 to 31 December 2020. In this study, we included only HDRs related to ordinary hospitalization, so we removed hospitalizations in day hospitals. We also removed HDRs of the year 2020 related to patients with a main or secondary diagnosis of COVID-19 (ICD9-CM 078.89 or ICD9-CM 078.89, respectively) and a main diagnosis of COVID-19-related disease. COVID-19-related diseases have been identified by guidelines enacted by the Italian Ministry of Health [21], and they are as follows: pneumonia in other infectious diseases classified elsewhere (ICD9-CM 484.8), acute bronchitis (ICD9-CM 466.0), bronchitis not specified as acute or chronic (ICD9-CM 490), other diseases of the respiratory system not elsewhere classified (ICD9-CM 519.8), acute respiratory failure (ICD9-CM 518.81), other pulmonary insufficiency not elsewhere classified (ICD9-CM 518.82), and fever (ICD9-CM 780.6).

The crude in-hospital mortality rate (IMR) was calculated as the ratio of the proportion of patients who died during hospitalization to the total number of hospitalized patients. The rates were calculated fortnightly by grouping the data accordingly, and they were standardized by the direct method, using the number of inpatient hospitalizations of Puglia in 2019 as a standard population. Standard rates are shown as the number of deaths per 10,000 inhabitants and its standard error (SE) value. The standardized rate ratios (SRRs) were calculated by dividing the 2020 standardized IMR by the 2019 standardized IMR, under the assumption that the population remained the same. Confidence intervals were calculated at the 95% confidence level under the assumption of normal distribution log (SRR).

To assess the possible effect of the pandemic on in-hospital mortality, the period under examination was divided into four periods, with decreasing levels of restriction:Phase 1, from 1 March 2020 to 30 April 2020, “total lockdown”, with a high level of restrictions: ban on leaving the house except out of necessity, suspension of educational services, closure of all commercial activities and public offices [6];Phase 2, from 1 May 2020 to 15 June 2020 and from 1 October 2020 to 31 December 2020, “soft lockdown,” with a moderate level of restriction: ban on moving out of town except for work, suspension of educational services, closure of some commercial activities [9,22];Phase 3, from 16 June 2020 to 30 September 2020, with a low level of restriction: suspension of educational indoor service, reduction of the number of people accessing commercial activities [23].Phase 4, from 1 January 2019 to 29 February 2020, with no restrictions except quarantine for COVID-19 patients.

To evaluate the effect of age at the time of hospitalization, patients were classified into 7 age groups: 0–4 years, 5–14 years, 15–24 years, 25–44 years, 45–64 years, 65–75 years, and >75 years. Comorbidities reported in secondary diagnoses were used to calculate the Charlson comorbidity index (CCI) score [24]. Clinical conditions and associated scores are defined by the Quan H. et al. [25] coding algorithm and are as follows: cerebrovascular disease, chronic lung disease, congestive heart failure, connective tissue disease, dementia, diabetes, mild liver disease, myocardial infarct, peripheral vascular disease, and ulcer were assigned 1 point each; any tumor, diabetes with organ damage, hemiplegia, leukemia, lymphoma, and moderate or severe kidney disease (eGFR < 60 mL/min) were assigned 2 points each; moderate or severe liver disease were assigned 3 points each; and AIDS and metastatic solid tumors were assigned 6 points each. CCI was then classified into three classes: CCI = 0, 1 ≤ CCI ≤ 2, and CCI ≥ 3. The length of stay (LoS) was also considered a possible modifier and was divided into three classes: 0–1 days, 2–5 days, and >5 days.

The possible association between in-hospital deaths, period of hospitalization, and covariates such as age class, sex, CCI class, and LoS were evaluated using a multivariable logistic regression model. The interactions between the phases and CCI class and that between the phases and LoS class were also included in the model. Results of the logistic regression model are shown as adjusted odds ratio for each variable. The odds ratio is the ratio between the risk of the outcome in a group with respect to the risk of the reference group. A positive association should be considered if the value of OR is higher than 1. Pairwise multiple comparisons were adjusted according to the Bonferroni correction. A *p*-value < 0.05 was considered statistically significant. Statistical analyses were performed using the SAS/STAT^®^ Statistics version 9.4(SAS Institute, Cary, NC, USA).

## 3. Results

The total number of HDRs included in this study was 782,704: 443,722 from 2019 and 338,982 from 2020. In total, 57,314 HDRs were excluded because they were not for ordinary hospitalizations. Further, 1000 HDRs from 2020 were excluded because they were related to patients with a main diagnosis of COVID-19 and 5030 because they were HDRs of secondary diagnosis of COVID-19 and main diagnosis of COVID-related diseases (Figure 1).

The mean age was 53.20 (SD 27.44); distribution for sex, age group, CCI class, length of stay class, and outcome (discharged dead or alive) are reported in Table 1.

### 3.1. Standardized Mortality Rates. Comparison between 2019 and 2020

In 2019, the number of deaths varied between 767 in the period from 16 January to 31 January and 464 in the period from 1 October to 15 October, while the highest standardized IMR was found between 1 January and 15 January (42.3 ± 1.6) and the lowest standardized IMR in the period between 16 October and 31 October (24.0 ± 1.1). In addition, the highest absolute number of in-hospital deaths (685) occurred in the second fortnight of January 2020, the same period as that of 2019, with the highest absolute number of events (767). The highest standardized IMRs occurred in the two periods between 16 March and 15 April, 56.8 ± 2.5 and 57.8 ± 2.7, respectively. In general, the standardized IMR in the year 2020 was higher than the 2019 IMR calculated every fortnight, except during the periods 15–30 January, 1–15 February, 16–29 February, and 1–15 July. A higher difference between 2020 and 2019 standard rates was found between 1 April and 15 April (IMR2019 25.2 ± 1.1; IMR2020 57.8 ± 2.7). All the in-hospital mortality standard rates for 2019 and 2020 are reported in Appendix A. In the period from 1 April to 15 April, we observed a higher SRR (SRR 2.29, 95% CI 2.02–2.60). The fortnightly periods from 1 March to 31 May and those from 15 October to 31 December have shown a sequence of SRRs significantly greater than one (Figure 2).

### 3.2. Effect of the Pandemic Phases on Mortality: Multivariable Model

A multivariable logistic regression model was applied to estimate the probability of in-hospital deaths related to the pandemic phases and other possible covariates such as age, sex, CCI class, and LoS class (Table 2). The risk of death was slightly higher in men than in women (OR 1.04, 95% CI: 1.01–1.07). Those in the lower age groups showed a lower risk compared to those in the adjacent younger age groups; higher risk of mortality was observed in younger patients in the 0–4 years age group, compared with those in the 5–14 years age group (OR 3.88, 95% CI 2.24–6.72). The evaluation of the period of restriction measures (phases) showed a greater risk of in-hospital mortality in the most critical phases of the pandemic, with the highest being observed in phase 1 compared to phase 3 and in phase 1 compared to phase 4. The difference between phases 3 and 4 was not statistically significant. The risk of in-hospital death was lower for hospitalizations with a lower CCI score. Regarding the effect of LoS, the risk of in-hospital death was higher on day 0–1 of hospitalization than in the other two LoS classes; in contrast, the risk was lower on days 2–5 of hospitalization than on day 6 or higher (OR 0.84, 95% CI 0.80–0.88).

Regarding the interaction between the pandemic phases and the CCI class (Figure 3), a higher risk was found in the phase 1 compared to phase 4, in the lowest severity class (CCI = 0) (OR 1.83, 95% CI 1.68–1.99). The odds ratios did not result as statistically significant: for patients with CCI = 0, phase 3 compared to phase 4 resulted in an OR of 1.02 (95% CI 0.96–1.09); for patients in the class 1 ≤ CCI ≤ 2, the risk in phase 3 compared to phase 4 had an OR of 1.06 (95% CI 0.99–1.12); for patients in the class CCI ≥3, the risk in phase 2 compared to phase 4, and that for phase 3 compared to phase 4 had odds ratios of 1.09 (95% CI 0.99–1.19) and 0.92 (95% CI 0.84–1.00), respectively.

The analysis of the interaction between the phases of the pandemic and LoS (Figure 4) showed that in the LoS classes of 0–1 day and 2–5 days, the ORs were all higher than 1, suggesting an increase of death risk for all the pairwise phases comparison. Moreover, given the lower confidence limit higher than 1, all odds ratios should be considered statistically significant with the exception for phase 1 compared with phase 2. In the LoS class of >5 days, the ORs of in-hospital mortality risk in phase 1 compared to phase 2 (OR 1.19, 95% CI 1.05–1.36), that of phase 1 compared to phase 3 (OR 1.26, 95% CI 1.12–1.41), and that of phase 1 compared to phase 4 (OR 1.37, 95% CI 1.24–1.49) were statistically significant. The other OR was not statistically significant (Figure 4).

Figure 5 shows the predicted probabilities of in-hospital mortality estimated through the multivariable logistic model for the different phases and at every level of interaction with the LoS and CCI classes. The probability of death was generally higher in short-term hospitalizations (0–1 day) and in patients with more comorbidities (CCI > 3). In phase 1, the probability was higher than that in phase 2 and, with the same LoS, this difference is greater for CCI ≥ 3. The difference in the predicted probability of death between phases 1 and 3 was higher for CCI > 0. The difference between phases 2 and 3 was more evident for LoS of 0–1 day and greater when CCI = 1–2. No difference was evident between phases 3 and 4.

## 4. Discussion

In this study, we examined the effects of the pandemic on non-COVID-19 patients in an area with over three million inhabitants. To the best of our knowledge, this is one of the few studies [15] that analyzed in-hospital non-COVID-19-related mortality trends to evaluate the short-term effects of the pandemic.

At the beginning of 2020 (from 16 January to 15 February), the 2020 standard IMR was lower than that of 2019, although the difference was not statistically significant. The same results were found in a Danish study [26], in which a reduction in overall mortality was reported. According to the authors of this study, it can be hypothesized that the reduction in mortality rate was due to the reduction of influenza cases (8,104,000 in 2019 against 7,595,000 cases in 2020 in Italy, with a population of 59.6 million people) [27]. This reduction could be due to the increase in the number of subjects who had completed the flu vaccination in the 2019–2020 vaccination campaign, compared with the 2018–2019 campaign (in Apulia region 17.8% in the 2019–2020 vs. 17% in the 2018–2019) [28].

The increase in IMR started when the lockdown began in Italy. Comparing the lockdown period (phase 1) to other restriction periods (phases 2, 3, and 4), we observed an increase in IMR independent of the comorbidities of the patients. The risk of death was higher in patients without comorbidity (CCI = 0) than in patients with comorbidity (1 ≤ CCI ≤ 2 and CCI ≥ 3). In a study [18] on the effect of public health restriction on heart diseases, an increase in IMR for myocardial infarction was observed in patients admitted during the lockdown period, as we have observed for all cases in this study. It is important to remember that, when the lockdown started, scheduled hospitalizations, with the exception of emergency hospitalization and oncological treatment, were all rescheduled. Nevertheless, in Italy, around the lockdown period, a reduction in the number of accesses to the ED was observed [15]. In the Apulia region, we observed an increase in the non-COVID-19-related IMR during the lockdown period. When restrictive measures of public health were slightly reduced, the non-COVID-19-related IMR returned to that of 2019, and the differences were not statistically significant. It is worth noting that after the new spread of the SARS-CoV-2 infection at the end of September, new sanitary restrictive measures started due to a new decree by the Ministry of Health [22], and an increase in IMR could be observed even if it was lower than that of the first lockdown. It is possible to speculate that people do not visit the hospital when they have a minor health problem (probably due to the public health restrictive strategy) but do when they have a complicated or serious health problem; in fact, we detected a reduction in hospitalization rates for those without COVID-19. Therefore, a delay in hospital access can be expected [18,19].

The total number of hospitalizations, in this study, for both medical and surgical diseases, was reduced in 2020 compared to 2019, probably due to the effect of the pandemic. In many studies conducted on EDs, a reduction in hospital access was observed, and as a consequence, a reduction of hospitalization rate [16,29]. In a study conducted on the EDs of the Lombardy region (the Italian region with the highest rate of SARS-CoV-2 infection), a severe reduction in access was observed from February 2020 to April 2020 and from October 2020 to December 2020. Despite the reduction in accesses to EDs, there was an increase in hospitalization rates and in the severity of patient access [30]. Interestingly, in the same period, we observed a major increase in IMR despite the Apulian diffusion curve of SARS-CoV-2 being lower than that of Lombardy region [31]. The fact that patients suffering from an acute pathology not related to COVID-19 did not visit the hospital due to the fear of contracting SARS-CoV-2 infection [32] and in accordance with the increasingly stringent lockdown measures cannot be ignored [26]. If people did not visit the hospital, it can be assumed that many people died in their homes, as suggested by a French study [33], in which it was estimated that the number of domestic deaths from cardiovascular diseases was equal to one-eighth of in-hospital deaths. Further studies on out-of-hospital mortality should be conducted to better analyze the trend of mortality in acute and chronic conditions during the pandemic.

In 2020, many countries worldwide implemented strategies to reduce the spread of SARS-CoV-2 infection and its consequences on hospital involvement. However, these strategies have had a dual effect: they reduce the spread of SARS-CoV-2, but they also have collateral effects on the development of other diseases [26]. In Italy, as well as in France and the United States, the public health restriction strategies determined a reduction in hospitalization rates, mostly for cardiovascular disease (coronary syndrome, atrial fibrillation, decompensated heart failure) and stroke [18,34]. Therefore, an excessive increase in mortality rate has been observed in many countries, which is caused directly and indirectly by the COVID-19 pandemic and the strategies implemented to mitigate it [35].

## 5. Study Limitation

This study had some limitations. Misclassification of diagnosis or incomplete fill-in of discharge records could have had an effect on mortality estimation. Furthermore, the short time window of our study does not allow us to effectively analyze the medium- and long-term effects of the pandemic on the use of health services and on the health outcome indicators such as in-hospital mortality. Moreover, this study analyzed the Apulia setting only; therefore, generalizations should be drawn with caution: Both the organization of the SSN and the degree of COVID-19 incidences were different among different geographical areas in Italy, as well as for wider geographical comparisons.

## 6. Conclusions

In summary, our analysis shows a possible association between in-hospital mortality in non-COVID-19-related diseases and restrictive measures of public health. The risk of in-hospital deaths increased during the lockdown period. In this period, it is conceivable that patients were hospitalized only for severe illnesses. Furthermore, access to the hospital appeared delayed, compared with that in previous years [17].

Based on what has been observed, we believe it is necessary to highlight that healthcare professionals must make targeted public health choices. These choices must be communicated correctly to people to prevent fear of contagion when they go to the hospital.

We also believe it is necessary to continue to conduct studies on the medium- and long-term effects of the pandemic on non-COVID patients, which can guide future public health choices.

## Figures and Tables

**Figure 1 ijerph-18-10886-f001:**
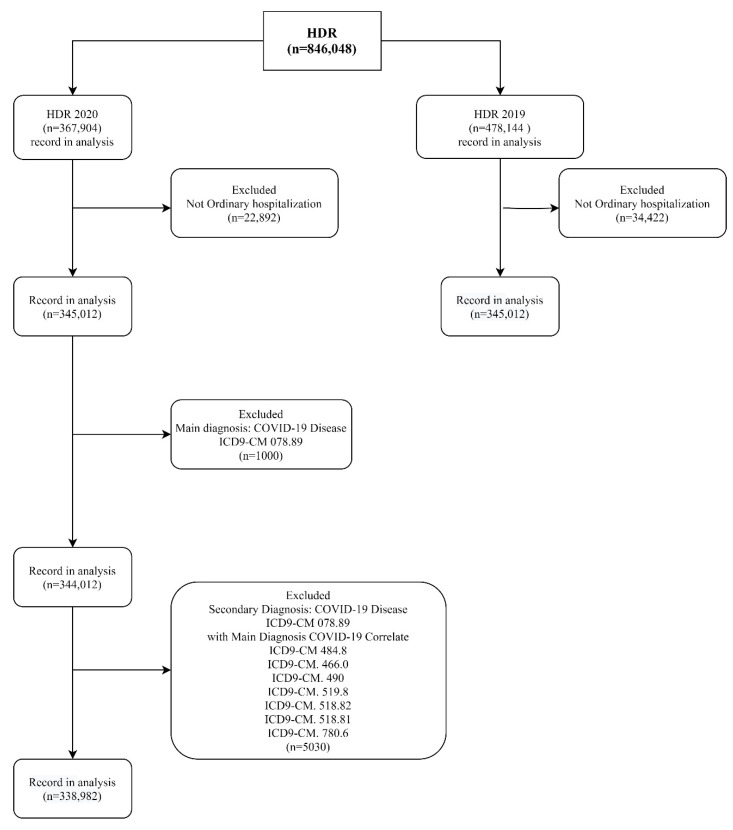
Flow chart of HDR (hospital discharge record) selection based on the inclusion and exclusion criteria. ICD 9 CM: International Code Disease, 9th revision, Clinical Modification.

**Figure 2 ijerph-18-10886-f002:**
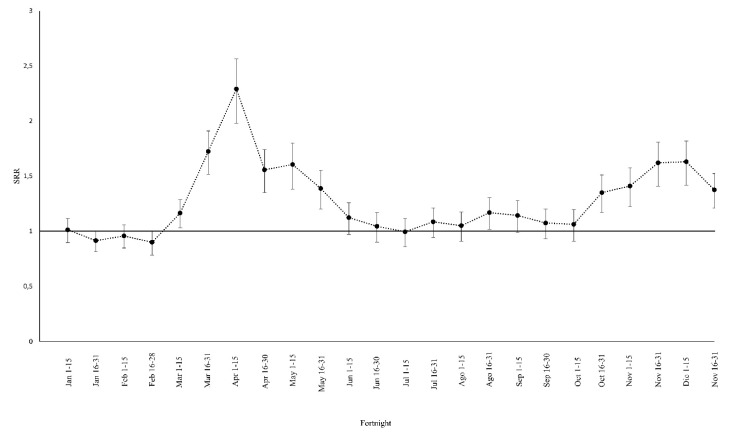
Standard rate ratio (SRR) and their 95% confidence interval between the years 2020 and 2019 for a fortnightly period. SSRs are statistically significant when the vertical bar of the confidence interval does not cross the horizontal line at the value 1.

**Figure 3 ijerph-18-10886-f003:**
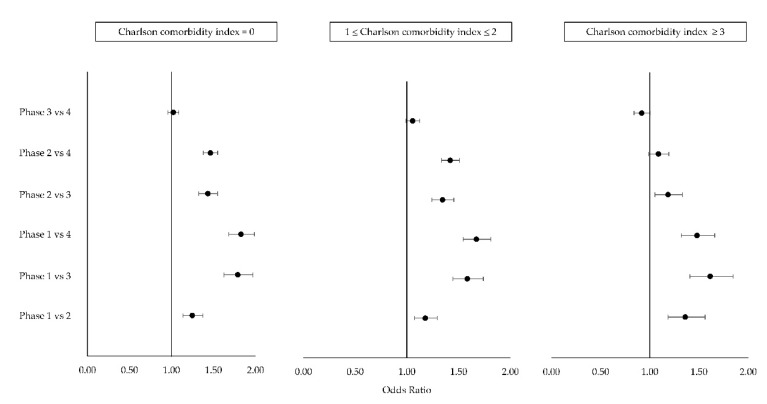
Forest plot of odds ratios and their 95% CI by pandemic phase and Charlson index. ORs are statistically significant when the horizontal bar of the confidence interval does not cross the vertical line at the value 1.

**Figure 4 ijerph-18-10886-f004:**
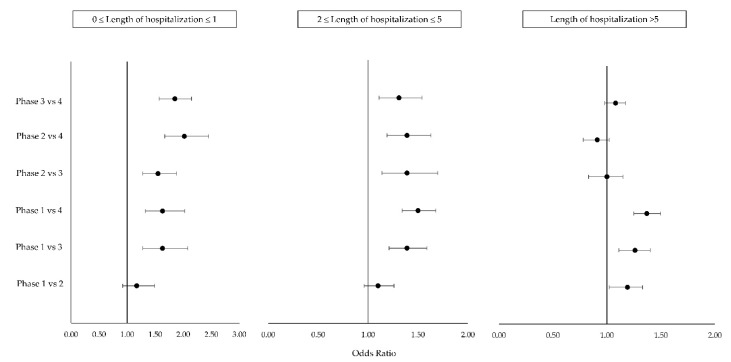
Forest plot of odds ratios and their 95% confidence intervals by pandemic phase and length of stay. ORs are statistically significant when the horizontal bar of the confidence interval does not cross the vertical line at the value 1.

**Figure 5 ijerph-18-10886-f005:**
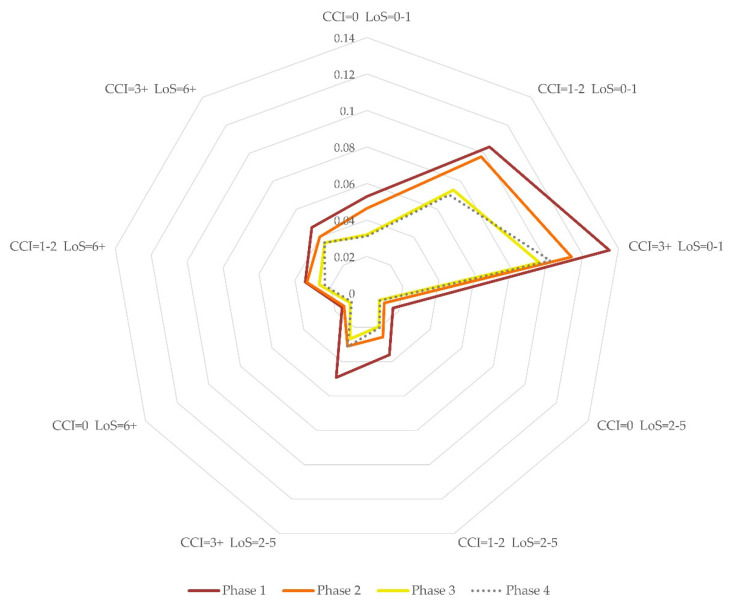
Individual predicted probability of in-hospital mortality by phase, length of stay (in the figure LoS) and Charlson index (CCI in the figure). Probabilities estimated for the mean effect of sex and age group.

**Table 1 ijerph-18-10886-t001:** Main characteristics of the hospital discharge records included in the study, by year of discharge.

Parameter	2019		2020		Total	
	n	%	n	%	n	%
Sex						
Male	214.274	48.3	162.698	48.0	376.972	48.2
Female	229.448	51.7	176.284	52.0	405.732	51.8
Age (years)						
0–4	46.034	10.4	36.346	10.7	82.380	10.5
5–14	14.680	3.3	8.353	2.5	23.033	2.9
15–24	18.218	4.1	12.756	3.8	30.974	4.0
25–44	70.994	16.0	56.757	16.7	127.751	16.3
45–64	98.052	22.1	75.082	22.2	173.134	22.1
65–74	76.613	17.3	60.572	17.9	137.185	17.5
>75	119.131	26.9	89.116	26.3	208.247	26.6
Hospital discharge						
Death	13.267	3.0	12.125	3.6	25.392	3.2
Alive	430.455	97.0	326.857	96.4	757.312	96.8
Charlson Index (score)						
0	316.421	71.3	242.920	71.7	559.341	71.5
1–2	95.568	21.5	72.402	21.4	167.970	21.5
≥3	31.733	7.2	23.660	7.0	55.393	7.1
Length of Stay (days)						
0–1	44.884	10.1	32.069	9.5	76.953	9.8
2–5	215.779	48.6	169.568	50.0	385.347	49.2
5+	183.059	41.3	137.345	40.5	320.404	40.9

**Table 2 ijerph-18-10886-t002:** Results of the multivariable logistic model applied to the probability of in-hospital death.

Parameter		Type 3 Effects		OR [CI 95%] ^1^
		Chi-Square	*p*-Value	
Sex	Male	7.8	0.0052	1.04 [1.01–1.07]
	Female			Ref.
Age (years)	0–4	10,199.1	<0.0001	0.04 [0.03–0.05]
	5–14			0.01 [0–0.02]
	15–24			0.03 [0.02–0.05]
	25–44			0.06 [0.05–0.07]
	45–64			0.23 [0.22–0.24]
	65–74			0.4 [0.38–0.42]
	>75			Ref.
Phase	1	397.4	<0.0001	1.65 [1.53–1.79]
	2			1.31 [1.24–1.39]
	3			1.00 [0.94–1.06]
	4			Ref.
Charlson Index (score)	0	2973.6	<0.0001	0.29 [0.27–0.31]
	1–2			0.71 [0.67–0.75]
	3+			Ref.
Length of Stay (days)	0–1	3196.5	<0.0001	3.47 [3.27–3.68]
	2–5			0.84 [0.8–0.88]
	5+			Ref.
Phase ∗ Ich class		40.3	<0.0001	see Figure 3
Phase ∗ LoS class		40.6	<0.0001	see Figure 4

^1^ Adjusted by Bonferroni method; OR, odds ratio; CI, confidence interval.

## Data Availability

Data sharing not applicable No new data were created or analyzed in this study. Data sharing is not applicable to this article.

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
