# Peer review of "In-Hospital Mortality in Non-COVID-19-Related Diseases before and during the Pandemic: A Regional Retrospective Study"

_ijerph, 2021, doi:10.3390/ijerph182010886_

Round 1
Reviewer 1 Report
The work is relevant to the time. However, there are many room to improve and result and its interpretation. The following are my suggestions.
1. I suggest providing a brief account of the significance of the study issue/debate in a plain language to make your paper appealing to read. Then link it with the Italian problems and current state of knowledge on the study issue. The following sentence requires writing in simple sentences, better in separate paragraph which would make readers easier to understand specific problems of the research. " Therefore, it could be argued that the reduction of patients’ access to hospital care had an effect on their health outcome, especially in cases of pathologies that already appeared related to promptness of care delivery". 2. How did you code male and female (male= 1 or female =1)? It would help to understand the result. 3. Please report the mean value of the odd ratio which makes readers easier to understand. It can be fitted on the table. 4. Your interpretation is too numerical/ technical which makes it complex for many readers with basic knowledge of statistics. For example, the length of stay 0-1 =3.47 means positive and 2-5 =0.84 means negative association as you presented odd ratio values. You wrote " in the LoS class of 0-1 day, ORs were statistically significant". You have not stated a positive relationship. It indicates you are also not knowledgeable about it. If you look at your results of "Analysis of Maximum likelihood estimate" table of the SAS output, you can understand them. But non-staticians cannot interpret that OR less than 1 is negatively associated and greater than 1 is positively associated. You are required to address the readers' problem to present and interpret the results. 5. It is not leveled what figures the vertical axis and horizontal axis mean for all figures. Are those figures correct and meaningful? You need to describe the way of interpreting the graphs in lay person words.Author Response
We have appreciated the suggestion to improve our work and you will find below a point by point answer with short explanation of what we have done, with the hope to reach the final approvation.
- I suggest providing a brief account of the significance of the study issue/debate in a plain language to make your paper appealing to read. Then link it with the Italian problems and current state of knowledge on the study issue. The following sentence requires writing in simple sentences, better in separate paragraph which would make readers easier to understand specific problems of the research. "Therefore, it could be argued that the reduction of patients’ access to hospital care had an effect on their health outcome, especially in cases of pathologies that already appeared related to promptness of care delivery".
Thanks for the suggestion. We have added short but essential informations on hospitalization trends, and we have modified the sentence and moved it before the evidence of other researches.
- How did you code male and female (male= 1 or female =1)? It would help to understand the result.
As shown in table 2, the "female" sex was used as a reference to determine the OR relating to the risk of death associated with the sex covariate. We have also added a sentence in the period preceding the table to better understand the result.
- Please report the mean value of the odd ratio which makes readers easier to understand. It can be fitted on the table.
Thanks for this comment that induce us to explain better the meaning of the odds ratio. In a multivariable logistic regression model we could have as a result individual predicted probability of the event or the risk of the event for classes. If we show all this results it could be not plain to read too numbers. Thus odds ratio summarized better results for classes, i.e. risk related to the characteristics of subjects. We have added, therefore, a sentence in the methods chapter that explain how to interpret the OR values.
- Your interpretation is too numerical/ technical which makes it complex for many readers with basic knowledge of statistics. For example, the length of stay 0-1 =3.47 means positive and 2-5 =0.84 means negative association as you presented odd ratio values. You wrote " in the LoS class of 0-1 day, ORs were statistically significant". You have not stated a positive relationship. It indicates you are also not knowledgeable about it. If you look at your results of "Analysis of Maximum likelihood estimate" table of the SAS output, you can understand them. But non-staticians cannot interpret that OR less than 1 is negatively associated and greater than 1 is positively associated. You are required to address the readers' problem to present and interpret the results.
As suggested by the reviewer, we have revised the results, clarified the meaning of the OR value by specifying better for which categories it is indicative of an increased risk of death
- It is not leveled what figures the vertical axis and horizontal axis mean for all figures. Are those figures correct and meaningful? You need to describe the way of interpreting the graphs in lay person words.
In the titles of Figures 2, 3 and 4 we specified that in addition to the estimate of the SRR, its 95% confidence interval is also graphically represented. We have also included a sentence that allows its interpretation regarding significance. We have inserted the title of the horizontal axis in Figures 3 and 4.
Best regards
Paolo Trerotoli
Reviewer 2 Report
Accept after minor revision, details in the attachment.

Author Response
We have appreciated the suggestions to improve our paper and we hope to have reach a better results in the report of our research. YOu will find below a point by point answer.
Abstract: The background of the problem is missing, the authors have repeatedly presented the purpose of the study, methods and so on.
As suggested by the reviewer we revised the abstract by adding the background that led to our study (and avoided repeating the purpose of the study).
Introduction: The authors describe in detail the pandemic period, and there is no
information about the hospitalization of patients before the pandemic time.
Thanks for the suggestion. We have added short but essential informations on hospitalization trends, and we have modified the sentence and moved it before the evidence of other researches.
The purpose of the study - it should be reformulated in a more clear and specific way, moreover, the purpose of the abstract and the main manuscript is different, it should be the same.
As suggested by the reviewer, we rephrased the purpose of the study in the introduction. Now there are no differences with the abstract.
In the methodology, please add information on how the data was accessed and how the data was protected, where and how were stored.
In the methodology we have added information on how the data was accessed and how the data was protected.
Descriptions of tables should be more precise, pleased describe below of the table all abbreviations.
We have added the descriptions of the abbreviations in the footnotes of the table
The list of references should be revised in line with the authors' guidelines
We have revised some references by adding the link for online access and the access date where they were missing.
Best regards
Paolo Trerotoli